# A Bounded Measure for Estimating the Benefit of Visualization (Part II): Case Studies and Empirical Evaluation

**DOI:** 10.3390/e24020282

**Published:** 2022-02-16

**Authors:** Min Chen, Alfie Abdul-Rahman, Deborah Silver, Mateu Sbert

**Affiliations:** 1Department of Engineering Science, University of Oxford, Oxford OX1 3QG, UK; 2Department of Informatics, King’s College London, London WC2R 2LS, UK; alfie.abdulrahman@kcl.ac.uk; 3Department of Electrical and Computer Engineering, Rutgers University, New Brunswick, NJ 08901, USA; silver@jove.rutgers.edu; 4Department of Informàtica i Matemàtica Aplicada, University of Girona, 17071 Girona, Spain; mateu@ima.udg.edu

**Keywords:** information theory, theory of visualization, cost–benefit analysis, divergence measure, benefit of visualization, human knowledge in visualization, abstraction, deformation, volume visualization, metro map

## Abstract

Many visual representations, such as volume-rendered images and metro maps, feature a noticeable amount of information loss due to a variety of many-to-one mappings. At a glance, there seem to be numerous opportunities for viewers to misinterpret the data being visualized, hence, undermining the benefits of these visual representations. In practice, there is little doubt that these visual representations are useful. The recently-proposed information-theoretic measure for analyzing the cost–benefit ratio of visualization processes can explain such usefulness experienced in practice and postulate that the viewers’ knowledge can reduce the potential distortion (e.g., misinterpretation) due to information loss. This suggests that viewers’ knowledge can be estimated by comparing the potential distortion without any knowledge and the actual distortion with some knowledge. However, the existing cost–benefit measure consists of an unbounded divergence term, making the numerical measurements difficult to interpret. This is the second part of a two-part paper, which aims to improve the existing cost–benefit measure. Part I of the paper provided a theoretical discourse about the problem of unboundedness, reported a conceptual analysis of nine candidate divergence measures for resolving the problem, and eliminated three from further consideration. In this Part II, we describe two groups of case studies for evaluating the remaining six candidate measures empirically. In particular, we obtained instance data for (i) supporting the evaluation of the remaining candidate measures and (ii) demonstrating their applicability in practical scenarios for estimating the cost–benefit of visualization processes as well as the impact of human knowledge in the processes. The real world data about visualization provides practical evidence for evaluating the usability and intuitiveness of the candidate measures. The combination of the conceptual analysis in Part I and the empirical evaluation in this part allows us to select the most appropriate bounded divergence measure for improving the existing cost–benefit measure.

## 1. Introduction

This two-part paper is concerned with the measurement of the benefit of visualization and viewers’ knowledge used in visualization. The history of measurement science shows that the development of measurements in different fields has not only stimulated scientific and technological advancements but also encountered some serious contentions due to instrumental, operational, and social conventions [1]. While the development of measurement systems, methods, and standards for visualization may take decades of research, one can easily imagine their impact on visualization as a scientific and technological subject.

“Measurement ... is defined as the assignment of numerals to objects or events according to rules” [2].

Rules may be defined based on physical laws (e.g., absolute zero temperature), observational instances (e.g., the freezing and boiling points of water), or social traditions (e.g., seven days per week). Without exception, measurement development in visualization aims to discover and define rules that will enable us to use mathematics in describing, differentiating, and explaining phenomena in visualization, as well as in predicting the impact of a design decision, diagnosing shortcomings in visual analytics workflows, and formulating solutions for improvement.

In 2016, Chen and Golan proposed an information-theoretic measure for quantifying the cost–benefit of visualization [3]. However, this measure consists of an unbounded divergence term, making the numerical measurements difficult to interpret. In the first part of this paper [4], Chen and Sbert:
Reviewed the related work that prepared for this cost–benefit measure, provided the measure with empirical evidence, and featured the application of the measure.Identified a shortcoming of using the Kullback–Leibler divergence (KL-divergence) in the cost–benefit measure and demonstrated the shortcoming using practical examples.Presented a theoretical discourse to justify the use of a bounded measure for finite alphabets.Proposed a new bounded divergence measure, while studying existing bounded divergence measures.Analyzed nine candidate measures using seven criteria reflecting desirable conceptual or mathematical properties, and narrowed the nine candidate measures to six measures.


In this second part of the paper, we focus on the remaining six candidate measures and evaluate them based on empirical evidence. In particular:


We report several case studies for collecting practical instances to evaluate the remaining candidate measures.We demonstrate the uses of the cost–benefit measurement to estimate the benefit of visualization in practical scenarios and the human knowledge used in the visualization processes.We report the discovery of a new conceptual criterion that a divergence measure is a summation of the entropic values of its components, which is useful in analyzing and visualizing empirical data.Finally, we bring the multi-criteria decision analysis (MCDA) in Parts I and II together and offer a recommendation to revise the information-theoretic measures proposed by Chen and Golan [3].


In addition, we use the data collected in two visualization case studies to explore the relationship between the benefit of visualization and the viewers’ knowledge used in visualization. As shown in Figure 1, in one case study, we asked participants to perform tasks for estimating the walking time (in minutes) between two underground stations indicated by a pair of red or blue arrows. Although the deformed London underground map was not designed to perform visualization tasks, many participants performed rather well, including those who had very limited experience of using the London underground. This suggests that with the presence of knowledge, a seemingly-tiny amount of visual information can be very useful.

We proposed two different ways of estimating viewers’ knowledge that has been used in the visualization process to alleviate the potential distortion. When we use different candidate measures to estimate viewers’ knowledge, we evaluate these candidate measures using the collected practical instances, while demonstrating that we are getting closer to be able to estimate the “benefit” of and “knowledge” used in practical visualization processes.

Readers are encouraged to consult the related reports on the cost–benefit analysis [5] and the Part I of this paper [4]. Nevertheless, this part of the paper is written in a self-contained manner.

## 2. Related Work

This two-part paper is concerned with information-theoretic measures for quantifying aspects of visualization, such as benefit, knowledge, and potential misinterpretation. The first part [4] focuses its review on previous information-theoretic work in visualization. In this section, we focus our review on previous measurement work in visualization.

### 2.1. Measurement Science

There is currently no standard measurement scale for measuring the benefit of visualization, levels of visual abstraction, the human knowledge used in visualization, or the potential to misinterpret visual abstraction. While these are considered to be complex undertakings, many scientists in the history of measurement science would have encountered similar challenges [1].

In their book [6], Boslaugh and Watters described measurement as “the process of systematically assigning numbers to objects and their properties, to facilitate the use of mathematics in studying and describing objects and their relationships.” They emphasized that measurement is not limited to physical qualities (e.g., height and weight) but also includes abstract properties (e.g., intelligence and aptitude). Pedhazur and Schmelkin [7] asserted the necessity of an integrated approach for measurement development, involving data collection, mathematical reasoning, technology innovation, and device engineering. Tal [8] pointed out that measurement is often not totally “real”, involves the representation of ideal systems and reflects conceptual, metaphysical, semantic, and epistemological understandings. Schlaudt [9] went one step further, referring measurement as a cultural technique.

This work is particularly inspired by the historical development of temperature scales and seismic magnitude. The former attracted the attention of many well-known scientists, benefited from both experimental observations (e.g., by Newton, Fahrenheit, Delisle, Celsius, etc.) and theoretical discoveries (e.g., by Boltzmann, Thomson (Kelvin), etc.). The latter started not long ago as the Richter scale was outlined in 1935. Since then, there have been many schemes proposed relating different physical properties. Many scales in both applications are related to logarithmic transformations.

Figure 2 depicted a number of instances that are quantified in different temperature scales. Isaac Newton proposed one of the first temperature scales based on his observation of over 20 instances [10]. Nine of them are shown in Figure 2, where the corresponding data in other scales were obtained based on Grigull’s study of the Netwon scale [11]. Although the Newton scale has not been adopted, his approach to mark and observe different data points at his proposed scale has been considered as “the first attempt to introduce an objective way of measuring ... temperature” [12].

From that first step, it took more than 40 years and many other proposals to developed the Celsius scale (with two of Newton’s data points as the reference points), which is most commonly used today. It took another century to develop the Kelvin scale with absolute zero as a new reference point. History motivates us to collected practical instances and conduct data-driven evaluation of candidate measures for estimating the benefit of visualization.

### 2.2. Metrics Development in Visualization

Behrisch et al. [13] presented a survey of quality metrics for information visualization. Bertini et al. [14] described a systemic approach of using quality metrics for evaluating high-dimensional data visualization focusing on scatter plots and parallel coordinates plots. A variety of quality metrics have been proposed to measure many different attributes, such as abstraction quality [15,16,17], quality of scatter plots [18,19,20,21,22,23,24], quality of parallel coordinates plots [25], cluttering [26,27,28], aesthetics [29], visual saliency [30], and color mapping [31,32,33,34].

In particular, Jänicke et al. [30] first considered a metric for estimating the amount of original data that is depicted by visualization and may be reconstructed by viewers. Chen and Golan [3] used the abstract form of this idea in defining their cost–benefit ratio. While the work by Jänicke et al. [30] relied on computer vision techniques for reconstruction, this work focused on collecting and analyzing empirical data because human knowledge has a major role to play in information reconstruction.

### 2.3. Measurement in Empirica Experiments

Almost all controlled empirical studies in visualization involve measuring the participants’ performance in visualization processes, typically in terms of accuracy and response time (e.g., [35]). Many uncontrolled empirical studies also collect participants’ experience and opinions qualitatively. Such collected data allow us to assess the benefit of visualization or potential misinterpretation. The empirical studies particularly relevant to this work are those on the topics of visual abstraction and human knowledge in visualization.

Isenberg [36] presented a survey of evaluation techniques on non-photorealistic and illustrative rendering. Isenberg et al. [37] reported an observational study comparing hand-drawn and computer-generated non-photorealistic rendering. Cole et al. [38] performed a study evaluating the effectiveness of line drawing in representing shape. Mandryk et al. [39] evaluated the emotional responses to non-photorealistic generated images. Liu and Li [40] presented an eye-tracking study examining the effectiveness and efficiency of schematic designs for depicting 30${}^{\circ }$ and 60${}^{\circ }$ directions in underground maps. Hong et al. [41] evaluated the usefulness of distance cartograms map “in the wild”. These studies confirmed that visualization users can deal with significant information loss due to visual abstraction in many situations.

Tam et al. [42] reported an observational study comparing automated and semi-automated machine learning (ML) workflows. Their information-theoretical analysis showed that ML developers entered a huge amount of knowledge (measured in bits) into a visualization-assisted ML workflow. Kijmongkolchai et al. [43] reported a study designed for detecting and measuring human knowledge used in visualization, and translated the traditional accuracy values to information-theoretic measures. They encountered an undesirable property of the Kullback–Leibler divergence in their calculations. In this work, we collect empirical data to evaluate the mathematical solutions proposed to address the issue encountered in [43].

If we can address this mathematical issue successfully, we will be able to complement qualitative methods for assessing the value of visualization (e.g., by Wall et al. [44]) with quantitative measurement; we will be able to carry out many experiments (e.g., those by Cleveland and McGill [35] and Saket et al. [45]) to examine the trade-off between alphabet compression and potential distortion [3]; we will be able to estimate the knowledge used (or gained) by the users in (or from) visualization as discussed by Sacha et al. [46]; and we will be able to transform the current qualitative methods for optimizing visual analytics workflow (e.g., [47]) to quantitative methods.

## 3. Overview, Notations, and Problem Statement

### 3.1. Brief Overview

Whilst hardly anyone in the visualization community would support any practice intended to deceive viewers, there have been many visualization techniques that inherently cause distortion to the original data. The deformed London underground map in Figure 1 shows such an example. The distortion in this example is largely caused by many-to-one mappings. A group of lines that would be shown in different lengths in a faithful map are now shown with the same length.

Another group of lines that would be shown with different geometric shapes are now shown as the same straight line. In terms of information theory, when the faithful map is transformed to the deformed map, a portion of information has been lost due to the many-to-one mappings. In this work, we follow the Shannon’s definition of information. Many-to-one mappings result in the reduction of Shannon entropy [48].

The common phrase that “the appropriateness of information loss depends on tasks” is not an invalid explanation. Partly by a similar conundrum in economics “what is the most appropriate resolution of time series for an economist”, Chen and Golan proposed an information-theoretic cost–benefit ratio for measuring various factors involved in visualization processes [3]. Its qualitative version is:(1)$$\frac{\mathrm{Benefit}}{\mathrm{Cos}t}=\frac{\mathrm{Alphabet}\;\mathrm{Compression}-\mathrm{Potential}\;\mathrm{Distortion}}{\mathrm{Cos}t}$$

This cost–benefit ratio was described and discussed in the first part of the paper [4]. Appendix A provides a more detailed explanation of this measure in the context of visualization, while Appendix B explains in detail how tasks and users are considered by this measure in the abstract. A more comprehensive introduction can be found in an arXiv report [5].

### 3.2. Mathematical Notations

Consider a simple metro map consisting of only two stations in Figure 3. We consider three different grid resolutions, with 1×1 cell, 2×2 cells, and 4×4 cells, respectively. The following set of rules determine whether a potential path is allowed or not:The positions of the two stations are fixed on each grid and there is only one path between the red station and the blue station.As shown on the top-right of Figure 3, only horizontal, and diagonal path-lines are allowed.When one path-line joins another, it can rotate by up to $\pm 45^{\circ }$.All joints of path-lines can only be placed on grid points.

For the first grid with the 1×1 cell, there is only one possible path. We define an alphabet A to contain this option as its only letter $a_{1}$, i.e., $A=\{a_{1}\}$. For the second grid with 2×2 cells, we have an alphabet $B=\{b_{1},b_{2},b_{3}\}$, consisting of three optional paths. For the third grid with 4×4 cells, there are 15 optional paths, which are letters of alphabet $C=\{c_{1},c_{2},\ldots ,c_{15}\}$. When the resolution of the grid increases, the alphabet of options becomes bigger quickly. We can imagine it gradually allows the designer to create a more faithful map.

To a designer of the underground map, at the 1×1 resolution, there is only one choice regardless of how much the designer would like to draw the path to reflect the actual geographical path of the metro line between these two stations. At the 2×2 and 4×4 resolutions, the designer has 3 and 15 options, respectively. Increasing the number of options is one factor that causes the increasing uncertainty about the selection of a specific option. The other factor is the provability of each option being selected. This *uncertainty* can be measured by Shannon entropy, which is defined as:$$H\left(Z\right)=-\sum_{i=1}^{n}p_{i}\log_{2}p_{i}\;\mathrm{where}\;p_{i}\in \left[0,1\right],\sum_{i=1}^{n}p_{i}=1$$
where Z is an alphabet, and can be replaced with A, B, or C. To calculate Shannon entropy, the alphabet Z needs to be accompanied by a *probability mass function* (PMF), which is written as P(Z). Each letter $z_{i}\in Z$ is thus associated with a probability value $p_{i}\in P$.

Note: In this paper, to simplify the notations in different contexts, for an information-theoretic measure, we use an alphabet Z and its PMF *P* interchangeably, e.g., H(P(Z))=H(P)=H(Z). Readers can find more mathematical background about information theory in [49] in general, and [5] in relation to this paper.

To ensure the calculation is easy to follow, we consider only the first two grids below. Let us first consider the single-letter alphabet A and its PMF *Q*. As n=1 and $q_{1}=1$, we have H(A)=0 bits. A is 100% certain, reflecting the fact that the designer has no choice.

The alphabet B has three design options $b_{1}$, $b_{2}$, and $b_{3}$. If they have an equal chance to be selected by the designer, we have a PMF $Q_{u}$ with $q_{1}=q_{2}=q_{3}=1/3$, and thus $H(Q_{u}\left(B\right))\approx 1.585$ bits. When we examine the three options in Figure 3, it is not unreasonable to consider a second scenario that the choice may be in favor of the straight line option $b_{1}$ in designing a metro map according to the real geographical data. If a different PMF $Q_{v}$ is given as $q_{1}=0.9,q_{2}=q_{3}=0.05$, we have $H(Q_{v}\left(B\right))\approx 0.569$ bits. The second scenario features less entropy and is thus of more certainty.

Consider that the designer is given a metro map designed using alphabet B, and is asked to produce a more abstract map using alphabet A. To the designer, it is a straightforward task, since there is only one option in A. When a group of viewers is visualizing the final design $a_{1}$, we could give these viewers a task to guess what may be the original map designed with B. If most viewers have no knowledge about the possible options $b_{2}$ and $b_{3}$, and almost all choose $b_{1}$ as the original design, we can describe their decisions using a PMF *P* such that $p_{1}=0.998,p_{2}=p_{3}=0.001$. Since *P* is not the same as either $Q_{u}$ or $Q_{v}$, the viewers’ decisions diverge from the actual PMF associated with B. This divergence can be measured using the Kullback–Leibler divergence (KL-divergence):$$D_{\mathrm{KL}}\left(P\left(Z\right)||Q\left(Z\right)\right)=\sum_{i=1}^{n}p_{i}\left(\log_{2}p_{i}-\log_{2}q_{i}\right)=\sum_{i=1}^{n}p_{i}\log_{2}\frac{p_{i}}{q_{i}}$$

Using $D_{\mathrm{KL}}$, we can calculate (i) if the original design alphabet B has the PMF $Q_{u}$, we have $D_{\mathrm{KL}}\left(P|\right|Q_{u})\approx 1.562$ bits; and (ii) if the original design alphabet B has the PMF $Q_{v}$, we have $D_{\mathrm{KL}}\left(P|\right|Q_{v})\approx 0.138$ bits. There is more divergence in case (i) than case (ii). Intuitively, we can guess this as *P* appears to be similar to $Q_{v}$.

Recall the qualitative formula in Equation (Equation 1). In the original mathematical definition [3], the benefit of a visual analytics process is defined as:(2)$$\mathrm{Benefit}=\mathrm{AC}-\mathrm{PD}=H\left(Z_{i}\right)-H\left(Z_{i+1}\right)-D_{\mathrm{KL}}(Z_{i}^{\prime }\left|\right|Z_{i})$$
where $Z_{i}$ is the input alphabet to the process and $Z_{i+1}$ is the output alphabet. $Z_{i}^{\prime }$ is an alphabet reconstructed based on $Z_{i+1}$. $Z_{i}^{\prime }$ has the same set of letters as $Z_{i}$ but likely a different PMF. In Equation (Equation 2), the first two terms, $H\left(Z_{i}\right)-H\left(Z_{i+1}\right)$, directly measure the amount of information loss in terms of Shannon entropy, while the third term, $D_{\mathrm{KL}}(Z_{i}^{\prime }\left|\right|Z_{i})$, measures the consequence of the information loss.

In terms of Equation (Equation 2), we have $Z_{i}=B$ with PMF $Q_{u}$ or $Q_{v}$, $Z_{i+1}=A$ with PMF *Q*, and $Z_{i}^{\prime }=B^{\prime }$ with PMF *P*. We can thus calculate the benefit in the two cases as:$$\begin{array}{cc} \mathrm{Benefit}\;\mathrm{of}\;\mathrm{case}\;\left(i\right) & =H\left(B\right)-H\left(A\right)-D_{\mathrm{KL}}(B^{\prime }\left||B\right)\\ \; & =H\left(Q_{u}\right)-H\left(Q\right)-D_{\mathrm{KL}}\left(P|\right|Q_{u})\\ \; & \approx 1.585-0-1.562=0.023\mathrm{bits} \end{array}$$
$$\begin{array}{cc} \mathrm{Benefit}\;\mathrm{of}\;\mathrm{case}\;\left(\mathrm{ii}\right) & =H\left(Q_{v}\right)-H\left(Q\right)-D_{\mathrm{KL}}\left(P|\right|Q_{v})\\ \; & \approx 0.569-0-0.138=0.431\mathrm{bits} \end{array}$$

In case (ii), because the viewers’ expectation is closer to the original PMF $Q_{v}$, there is more benefit in the visualization process than case (i) though case (ii) has less AC than case (i).

However, $D_{\mathrm{KL}}$ has an undesirable mathematical property. If we consider a third case, (iii), where the original PMF $Q_{w}$ is strongly in favor of $b_{2}$, such as $q_{1}=\epsilon ,q_{2}=1-2\epsilon ,q_{3}=\epsilon$, where 0<ϵ<1 is a small positive value. If ϵ=0.001, $D_{\mathrm{KL}}\left(P|\right|Q_{w})=9.933$ bits. If ϵ→0, $D_{\mathrm{KL}}\left(P|\right|Q_{w})\to \infty$. Since the maximum entropy (uncertainty) for B is only about 1.585 bits, it is difficult to interpret that viewers’ divergence can be more than that maximum, not to mention the infinity.

### 3.3. Problem Statement

When using $D_{\mathrm{KL}}$ in Equation (Equation 1) in a relative or qualitative context (e.g., [47,50]), the unboundedness of the KL-divergence does not pose an issue. However, this does become an issue when $D_{\mathrm{KL}}$ is used to measure the PD in an absolute and quantitative context.

In the first part of this paper [4], Chen and Sbert showed that, conceptually, it is the unboundedness that is not consistent with a conceptual interpretation of KL-divergence for measuring the inefficiency of a code (alphabet) that has a finite number of codewords (letters). They proposed to find a suitable bounded divergence measure to replace the $D_{\mathrm{KL}}$ term in Equation (Equation 2). They examined nine candidate measures, analyzed their mathematical properties with the aid of visualization, and narrowed these down to six measures using multi-criteria decision analysis (MCDA) [51].

In this work, we continue their MCDA process by introducing criteria based on the analysis of instances obtained when using the remaining six candidate measures in different case studies, which correspond to criteria $S_{1}$, $S_{2}$, $R_{1}$, and $R_{2}$ in Table 1 that is presented in Section 7.

For self-containment, we give the mathematical definition of the six candidate measures below. In this second part of the paper, we treat them as black-box functions, since they have already undergone the conceptual evaluation in the first part of this paper. For more detailed conceptual and mathematical discourse on these six candidate measures, please consult that part [4].

The first candidate measure is Jensen–Shannon divergence [52], which is defined as:(3)$$\begin{array}{cc} D_{\mathrm{JS}}\left(P||Q\right) & =\frac{1}{2}\left(D_{\mathrm{KL}}\left(P||M\right)+D_{\mathrm{KL}}\left(Q||M\right)\right)=D_{\mathrm{JS}}\left(Q||P\right)\\ \; & =\frac{1}{2}\sum_{i=1}^{n}\left(p_{i}\log_{2}\frac{2p_{i}}{p_{i}+q_{i}}+q_{i}\log_{2}\frac{2q_{i}}{p_{i}+q_{i}}\right) \end{array}$$
where *P* and *Q* are two PMFs associated with the same alphabet Z and *M* is the average distribution of *P* and *Q*. Each letter $z_{i}\in Z$ is associated with a probability value $p_{i}\in P$ and another $q_{i}\in Q$. With the base 2 logarithm as in Equation (Equation 3), $D_{\mathrm{JS}}\left(P||Q\right)$ is bounded by 0 and 1.

The second candidate measure is the square root of $D_{\mathrm{JS}}$. The conceptual evaluation gave both $D_{\mathrm{JS}}$ and $\sqrt{D_{\mathrm{JS}}}$ the same promising score 30 as shown in Table 1. The third and fourth candidate measures are two instances of a new measure $D_{\mathrm{new}}^{k}$ proposed by Chen and Sbert [4]. The two instances are $D_{\mathrm{new}}^{k}\left(k=1\right)$ and $D_{\mathrm{new}}^{k}\left(k=2\right)$. They received scores of 28 and 30, respectively, in the conceptual evaluation. $D_{\mathrm{new}}^{k}$ is defined as follows: (4)$$D_{\mathrm{new}}^{k}\left(P||Q\right)=\frac{1}{2}\sum_{i=1}^{n}\left(p_{i}+q_{i}\right)\log_{2}\left(|p_{i}-q_{i}{|}^{k}+1\right)$$
where k>0. As $0\leq |p_{i}-q_{i}{|}^{k}\leq 1$, we have
$$\frac{1}{2}\sum_{i=1}^{n}\left(p_{i}+q_{i}\right)\log_{2}\left(0+1\right)\leq D_{\mathrm{new}}^{k}\left(P||Q\right)\leq \frac{1}{2}\sum_{i=1}^{n}\left(p_{i}+q_{i}\right)\log_{2}\left(1+1\right)$$

Since $\log_{2}1=0$, $\log_{2}2=1$, $\sum p_{i}=1$, $\sum q_{i}=1$, $D_{\mathrm{new}}^{k}\left(P||Q\right)$ is thus bounded by 0 and 1.

The fifth and sixth candidate measures are two instances of a non-commutative version of $D_{\mathrm{new}}^{k}$. It is denoted as $D_{\mathrm{ncm}}^{k}$, and the two instances are $D_{\mathrm{ncm}}^{k}\left(k=1\right)$ and $D_{\mathrm{ncm}}^{k}\left(k=2\right)$, which also received scores of 26 and 29, respectively, in the conceptual evaluation. $D_{\mathrm{ncm}}^{k}$ is defined as follows: (5)$$D_{\mathrm{ncm}}^{k}\left(P||Q\right)=\sum_{i=1}^{n}p_{i}\log_{2}\left(|p_{i}-q_{i}{|}^{k}+1\right),$$
which captures the non-commutative property of $D_{\mathrm{KL}}$.

As $D_{\mathrm{JS}}$, $\sqrt{D_{\mathrm{JS}}}$, $D_{\mathrm{new}}^{k}$, and $D_{\mathrm{ncm}}^{k}$ are bounded by [0, 1], if any of them is selected to replace $D_{\mathrm{KL}}$, Equation (Equation 2) can be rewritten as
(6)$$\mathrm{Benefit}=H\left(Z_{i}\right)-H\left(Z_{i+1}\right)-H_{\max}\left(Z_{i}\right)D(Z_{i}^{\prime }\left|\right|Z_{i})$$
where $H_{\max}$ denotes maximum entropy, while D is a placeholder for $D_{\mathrm{JS}}$, $\sqrt{D_{\mathrm{JS}}}$, $D_{\mathrm{new}}^{k}$, or $D_{\mathrm{ncm}}^{k}$. Note that while $H_{\max}\left(Z_{i}\right)D(Z_{i}^{\prime }\left|\right|Z_{i})$ is bounded by $H_{\max}\left(Z_{i}\right)$, $H_{\max}\left(Z_{i}\right)$ can have any non-negative value and is calculated as $\log_{2}∥Z_{i}∥$, where $∥Z_{i}∥$ is the number of letters in $Z_{i}$.

## 4. Evaluation Methodology and Criteria

Historically, developing different temperature scales is motivated by the need for defining and quantifying the divergence between any pair of values representing two instances of different temperatures. Isaac Newton approached this problem by collecting over 20 instances, nine of which are shown in Figure 2.

Given two PMFs *P* and *Q* associated with an alphabet Z with *n* letters, measuring the divergence between *P* and *Q* involves the definition and quantification of the interaction between *n* pairs of probability values. Hence, the measuring function is *n*-dimensional and is likely more difficult to define. Nevertheless, we can adapt Isaac Newton’s approach of using data points with practical meanings. Unlike the Newton scale, we do not need these data points to specify a scale, but only to evaluate candidate measures. Analogously, this is similar to use Newton’s data points to evaluate other temperature scales in Figure 2.

Consider a real world phenomenon being visualized by a user or a group of users for a specific task. Let alphabet $Z_{w}$, $Z_{v}$, and $Z_{t}$ be the information spaces of the phenomenon, the visualization, and the task concerned, respectively. There are two major transformations, one from $Z_{w}$ to $Z_{v}$ and another from $Z_{v}$ to $Z_{t}$. The first major transformation may contain processes for data capture, data processing, and data visualization, while the second major transformation may contain all cognitive processes from viewing to task performance. Both transformations may feature alphabet compression and potential distortion.

Most visualization tasks (including confirmation, categorization, recognition, search, estimation, etc.) can be abstracted as a decision to select from two or more options. A decision alphabet $Z_{t}$ essentially contains all valid options with a PMF. In some cases, there can be numerous options (e.g., counting). There is a ground truth PMF *Q* that reflects the ideal task performance when users have full access to perfect data sampled in the information space of the phenomenon $Z_{w}$, have an infinite amount of time to view the data with or without visualization, and do not have any cognitive bias in selecting the correct option. Although an accurate *Q* may be difficult to obtain, one can estimate it in a synthetic or real-world case study, which will be demonstrated in the next two sections.

As the transformation from $Z_{w}$ to $Z_{v}$ will lose a fair amount of information, users, who have a different amount of knowledge about the phenomenon, the visual representation, and the task concerned, will perform differently. Such difference will be captured in the PMF, *P*, compiled according to the actual task performance. or example, consider three typecasting cases: (a)*P is close to a uniform PMF*$P_{\mathrm{uniform}}$, *while the ground truth Q is dissimilar to a uniform PMF*—This suggests that the users may not have adequate knowledge and may have been making random guesses. In such a case, their task performance would lead to a PMF similar to $P_{\mathrm{uniform}}$.(b)*P is close to a PMF*$P_{\mathrm{visinfo}}$*that characterizes the available visual information while the ground truth Q differs from*$P_{\mathrm{visinfo}}$*noticeably*—This suggests that the users may not have adequate knowledge and may have been reasoning about the options in $Z_{t}$ entirely based on what is depicted visually. In such a case, their performance would result in a PMF similar to $P_{\mathrm{visinfo}}$.(c)*P is close to the ground truth Q, while Q differs from*$P_{\mathrm{uniform}}$*and*$P_{\mathrm{visinfo}}$*noticeably*—This suggests that the users may have been able to make the perfect combination of the available visual information and their knowledge. In such a case, their task performance could lead to a PMF similar to the ideal PMF *Q*.

One obvious method to determine whether a visual design is suitable for a group of users is to ask these users to perform some tasks. For a particular task, the users’ task performance can be sampled and approximated using a PMF $P_{\mathrm{sampled}}$. All the candidate measures can quantify the divergence between $P_{\mathrm{sampled}}$ and the ground truth PMF *Q*. We can also use such a candidate measure $D_{*}$ to quantify the benefit of visualization as: $$\mathrm{Benefit}=H\left(Q\right)-H\left(P_{\mathrm{visinfo}}\right)-H_{\max}\left(Q\right)D_{*}\left(P_{\mathrm{sampled}}∥Q\right)$$

Before we are able to reach the final conclusion, we consider that $D_{*}$ may be any one of the six candidate measures given in Section 3.3, i.e., $D_{\mathrm{JS}}$, $\sqrt{D_{\mathrm{JS}}}$, $D_{\mathrm{new}}^{k=1}$, $D_{\mathrm{new}}^{k=2}$, $D_{\mathrm{ncm}}^{k=1}$, and $D_{\mathrm{ncm}}^{k=2}$. In addition, we can also estimate the impact of the human knowledge used in performing a visualization task as: $$\begin{array}{c} K_{\upsilon }=H_{\max}\left(Q\right)\left(D_{*}\left(P_{\mathrm{visinfo}}∥Q\right)-D_{*}\left(P_{\mathrm{sampled}}∥Q\right)\right)\\ K_{\psi }=H_{\max}\left(Q\right)\left(D_{*}\left(P_{\mathrm{uniform}}∥Q\right)-D_{*}\left(P_{\mathrm{sampled}}∥Q\right)\right) \end{array}$$
where $K_{\upsilon }$ is an estimation against the scenario where users rely only on visual information without using any knowledge, and $K_{\psi }$ is that against the scenario of random guesses. If $K_{\upsilon }>0$ and $K_{\psi }>0$, they suggest a positive impact of human knowledge. If $K_{\upsilon }<0$ or $K_{\psi }<0$, they suggest some biases.

Given some instance data in the form of PMFs $P_{\mathrm{sampled}}$, we would like to observe how different candidate measures would (i) order these instances in terms of their divergence against an estimated ground truth PMF *Q*, (ii) quantify the benefit at a scale meaningful to visualization scientists, and (iii) assign the sensible signs to $K_{\upsilon }$ and $K_{\psi }$.

## 5. Synthetic Case Studies

We first consider two synthetic case studies, $S_{1}$ and $S_{2}$, which allow us to define idealized situations, from which collected data do not contain any noise. In many ways, this is similar to testing a piece of software using pre-defined test cases. Nevertheless, these test cases feature more complex alphabets than those considered by the conceptual evaluation presented in the first part of this paper [4].

### 5.1. Synthetic Case S_1_

Let $Z_{w}$ be a phenomenon alphabet with two letters, *good* and *bad*, for describing a scenario (e.g., an object or an event), where the ground truth probability of *good* is $q_{1}=0.8$, and that of *bad* is $q_{2}=0.2$. In other words, Q={0.8,0.2}. Imagine that a biased process (e.g., a distorted visualization, faulty data collection, an incorrect algorithm, or a misleading communication) conveys the information about the scenario always *bad*, i.e., a visualization alphabet $Z_{v}$ with a PMF $P_{\mathrm{visinfo}}=R_{\mathrm{biased}}=\left\{0,1\right\}$. Users at the receiving end of the process may have different knowledge about the actual scenario, and they will make a decision, $Z_{t}$, after receiving the output of the process. For example, there are five users, and we obtained the probability of their decisions (with different $P_{\mathrm{sampled}}$) as follows:LD—The user has a little doubt about the output of the process, and decides the letter of *bad* 90% of the time, and the letter of *good* 10% of the time, i.e., with PMF $P_{\mathrm{LD}}=\left\{0.1,0.9\right\}$.FD—The user has a fair amount of doubt, with $P_{\mathrm{FD}}=\left\{0.3,0.7\right\}$.RG—The user makes a random guess, with $P_{\mathrm{RG}}=\left\{0.5,0.5\right\}$.UC—The user has adequate knowledge about $Z_{w}$ but under-compensates it slightly, with $P_{\mathrm{UC}}=\left\{0.7,0.3\right\}$.OC—The user has adequate knowledge about $Z_{w}$ but over-compensates it slightly, with $P_{\mathrm{OC}}=\left\{0.9,0.1\right\}$.

We can use different candidate measures to compute the divergence between *Q* and each $P_{\mathrm{sampled}}$. The bar chart in Figure 4 shows different divergence values returned by these measures, while the transformations from *Q* to $R_{\mathrm{biased}}$ and then to different $P_{\mathrm{sampled}}$ are illustrated on the right margin of the figure. Each value is decomposed into two parts, one for *good* and one for *bad*, except that the candidate measure $\sqrt{D_{\mathrm{JS}}}$ cannot distinguish the component measures for individual letters since it is a global transformation after $D_{\mathrm{JS}}$ is calculated. This shortcoming of $\sqrt{D_{\mathrm{JS}}}$ was not noticed in the conceptual analysis in the Part I of this paper [4].

All these measures can order these five users reasonably well. The users UC (under-compensate) and OC (over-compensate) have the same values with $D_{\mathrm{new}}^{k}$ and $D_{\mathrm{ncm}}^{k}$, while $D_{\mathrm{JS}}$ and $\sqrt{D_{\mathrm{JS}}}$ consider OC has slightly more divergence than UC. For $D_{\mathrm{JS}}$, UC:OC = 0.010:0.014 and for $\sqrt{D_{\mathrm{JS}}}$, UC:OC = 0.098:0.120.

$D_{\mathrm{ncm}}^{k=1}$ and $D_{\mathrm{ncm}}^{k=2}$ show strong asymmetric patterns between *good* and *bad*, reflecting the probability values in $P_{\mathrm{sampled}}$. In other words, the more decisions on *good*, the more *good*-related divergence. This asymmetric pattern is not in any way incorrect, as the KL-divergence is also non-commutative and would also produce much stronger asymmetric patterns. An argument for supporting commutative measures would point out that the higher probability of *good* in *Q* should also influence the balance between the *good*-related divergence. We are slightly in favor of commutativity as it is easier to interpret. In terms of **ordering**, we consider $D_{\mathrm{new}}^{k}$ “excellent”, $D_{\mathrm{ncm}}^{k}$ “good” due to asymmetry, and $D_{\mathrm{JS}}$ and $\sqrt{D_{\mathrm{JS}}}$ “adequate” as the non-equal UC and OU measures are not so intuitive.

As H(Q)=0.722 and $H(R_{\mathrm{biased}})=0$, the amount of alphabet compression (AD) is 0.722 bits. $H_{\max}\left(Q\right)=1$ bit. We can compute the benefits of the visualization to the six users, which are shown in the left parallel coordinate plot (PCP) in Figure 4. From these PCPs, we notice that $D_{\mathrm{JS}}$, $\sqrt{D_{\mathrm{JS}}}$, $D_{\mathrm{ncm}}^{k=2}$, and $D_{\mathrm{new}}^{k=2}$ give positive benefits to all five users, with $D_{\mathrm{JS}}$ returning the highest values. $D_{\mathrm{ncm}}^{k=1}$, and $D_{\mathrm{new}}^{k=1}$ yield negative benefit values for user LD, which is consistent with our expectation. In terms of **benefit quantification**, we consider $D_{\mathrm{ncm}}^{k=1}$ and $D_{\mathrm{new}}^{k=1}$ are “excellent”, $\sqrt{D_{\mathrm{JS}}}$, $D_{\mathrm{ncm}}^{k=2}$, and $D_{\mathrm{new}}^{k=2}$ “good”; and $D_{\mathrm{JS}}$ “adequate”.

With $P_{\mathrm{visinfo}}=\left\{0,1\right\}$ and $P_{\mathrm{uniform}}=\left\{0.5,0.5\right\}$, we can calculate $K_{\upsilon }$ and $K_{\psi }$, which are shown in the middle and right PCPs in Figure 4, respectively. From the two PCPs, we cannot observe any major issue in categorizing positive and negative impact by any candidate measure. Hence, for **knowledge impact**, we consider all “excellent”.

To continue the multi-criteria decision analysis (MCDA) [51] in the first part of the paper [4], we rate the six candidate measures using the same scoring system, i.e., using ordinal values between 0 and 5 (0 unacceptable, 1 fall-short, 2 inadequate, 3 mediocre, 4 good, and 5 best). By combining our evaluation of the ordering of divergence, benefit quantification, and the sign of knowledge impact, we give a 5 score to $D_{\mathrm{new}}^{k=1}$, a 4 to $D_{\mathrm{new}}^{k=2}$ and $D_{\mathrm{ncm}}^{k=1}$, a 3 to $D_{\mathrm{ncm}}^{k=2}$, a 2 to $\sqrt{D_{\mathrm{JS}}}$ and a 1 to $D_{\mathrm{JS}}$. The qualitative rating and numerical scores are given in Table 1.

### 5.2. Synthetic Case S${}_{2}$

We now consider a slightly more complicated scenario with four pieces of data, A, B, C, and D, which can be defined as an alphabet $Z_{w}$ with four letters. The ground truth PMF is Q={0.1,0.4,0.2,0.3}. Consider two processes that combine these into two classes AB and CD, each resulting in a two-letter alphabet $Z_{v}$. These typify clustering algorithms, downsampling processes, discretization in visual mapping, and so on. One process is considered to be *correct*, which has a PMF for AB and CD as $R_{\mathrm{correct}}=\left\{0.5,0.5\right\}$, and another *biased* process with $R_{\mathrm{biased}}=\left\{0,1\right\}$.

Let CG, CU, and CB be three users at the receiving end of the *correct* process, and BG, BS, and BM be three other users at the receiving end of the *biased* process. The users with different knowledge exhibit different abilities to reconstruct the original $Z_{w}$ featuring A, B, C, and D from aggregated information about AB and CD in $Z_{v}$. Similar to the *good*-*bad* scenario, such abilities can be captured by a PMF $P_{\mathrm{sampled}}$. For example, we have:CG makes random guess, $P_{\mathrm{CG}}=\left\{0.25,0.25,0.25,0.25\right\}$.CU has useful knowledge, $P_{\mathrm{CU}}=\left\{0.1,0.4,0.1,0.4\right\}$.CB is highly biased, $P_{\mathrm{CB}}=\left\{0.4,0.1,0.4,0.1\right\}$.BG makes guess based on $R_{\mathrm{biased}}$, $P_{\mathrm{BG}}=\left\{0.0,0.0,0.5,0.5\right\}$.BS makes a small adjustment, $P_{\mathrm{BS}}=\left\{0.1,0.1,0.4,0.4\right\}$.BM makes a major adjustment, $P_{\mathrm{BM}}=\left\{0.2,0.2,0.3,0.3\right\}$.

Figure 5 compares the divergence values returned by the candidate measures for these six users, while the transformations from *Q* to $R_{\mathrm{correct}}$ or $R_{\mathrm{biased}}$, and then to $P_{\mathrm{sampled}}$ are illustrated on the right. Different measures provided slightly different ordering of the six users as: $$\begin{array}{cc} D_{\mathrm{JS}},\sqrt{D_{\mathrm{JS}}},D_{\mathrm{new}}^{k=2}: & \mathrm{CU}<\mathrm{CG}<\mathrm{BM}<\mathrm{BS}<\mathrm{CB}<\mathrm{BG}\\ D_{\mathrm{ncm}}^{k=2}: & \mathrm{CU}<\mathrm{CG}<\mathrm{BM}<\mathrm{BS}<[\mathrm{CB},\mathrm{BG}]\\ D_{\mathrm{ncm}}^{k=1}: & \mathrm{CU}<\mathrm{BM}<\mathrm{CG}<\mathrm{BS}<[\mathrm{CB},\mathrm{BG}]\\ D_{\mathrm{new}}^{k=1}: & \mathrm{CU}<\mathrm{BM}<\mathrm{CG}<\mathrm{BS}<\mathrm{CB}<\mathrm{BG}\\ \mathrm{collective}\;\mathrm{votes}: & \mathrm{CU}<\mathrm{CG}<\mathrm{BM}<\mathrm{BS}<\mathrm{CB}<\mathrm{BG} \end{array}$$

The **order of divergence** can be observed in the bar chart as the first PCP where the divergence values are scaled by $H_{\max}=2$ bits. Using the collective votes as the benchmark, we consider $D_{\mathrm{JS}}$, $\sqrt{D_{\mathrm{JS}}}$, and $D_{\mathrm{new}}^{k=2}$ “excellent”, $D_{\mathrm{ncm}}^{k=2}$ and $D_{\mathrm{new}}^{k=1}$ “good”, and $D_{\mathrm{ncm}}^{k=1}$ “adequate”.

The PCPs in Figure 5 also depict two additional sets of values for $P_{\mathrm{visinfo}}$ when a user relies solely on visual information. For CG, CU, and CB, the benchmark is Cvi that corresponds to $R_{\mathrm{corrected}}$. For BG, BS, and BM, the benchmark is Bvi that corresponds to $R_{\mathrm{biased}}$. From the first PCP, we can observe that Bvi causes more distortion than Cvi.

However, because the entropy of the ground truth alphabet H(Q)=1.84, and the entropy values of $R_{\mathrm{corrected}}$ and $R_{\mathrm{biased}}$ are 1 and 0 bits, $R_{\mathrm{biased}}$ results in more alphabet compression. The second PCP shows that if a user relies solely on visual information, $R_{\mathrm{biased}}$ leads to more benefit, except that $D_{\mathrm{ncm}}^{k=2}$ thinks otherwise. We cannot find major issues with other benefit values in the second PCP, though we consider that the negative values produced by $\sqrt{D_{\mathrm{JS}}}$, $D_{\mathrm{ncm}}^{k=1}$, and $D_{\mathrm{new}}^{k=1}$ are intuitive. In terms of **benefit quantification**, we consider $\sqrt{D_{\mathrm{JS}}}$ to be “excellent”, $D_{\mathrm{ncm}}^{k=1}$ and $D_{\mathrm{new}}^{k=1}$ “good”, and the others “adequate”.

Observing the third and fourth PCPs is interesting. The clustering algorithm changes Q={0.1,0.4,0.2,0.3} to $R_{\mathrm{correct}}=\left\{0.5,0.5\right\}$ for users CG, CU, and CB and $R_{\mathrm{biased}}=\left\{0,1\right\}$ for users BG, BS, and BM. The random guess of *Q* with a uniform distribution is not that bad. Only CU’s knowledge has a positive impact against random guess as shown in the last PCP.

Against the less-ideal visual information characterized by $R_{\mathrm{correct}}$ and $R_{\mathrm{biased}}$, the knowledge of all six users has a positive impact. It is important to state here the **knowledge** can be gained from other visualization. For example, we can postulate that the reason CU, BS, and BM can make adjustments against what the clustering algorithm says is because they have seen some visualizations of the raw data without clustering at an early stage of a workflow. In general, we cannot find any major issues with the PCPs for $K_{\upsilon }$ and $K_{\psi }$. We thus rate all candidate measures as “excellent”.

By combining our evaluation of the ordering of divergence, benefit quantification, and the sign of knowledge impact, we give a 5 score to $\sqrt{D_{\mathrm{JS}}}$, a 3 to $D_{\mathrm{JS}}$, $D_{\mathrm{new}}^{k=1}$ and $D_{\mathrm{new}}^{k=2}$, and a 2 to $D_{\mathrm{ncm}}^{k=1}$ and $D_{\mathrm{ncm}}^{k=2}$.

### 5.3. An Extra Conceptual Criterion

The square root of JS-divergence, i.e., $\sqrt{D_{\mathrm{JS}}}$, is the only candidate measure that is not the probabilistic mean of its component measures, which correspond to the letters of the alphabet concerned. From the perspective of visualization, it cannot be depicted in the same way as the other five measures. As demonstrated in Figure 4 and Figure 5, $\sqrt{D_{\mathrm{JS}}}$ values are depicted in grey bars, and one cannot view the individual contributions of its components to the overall divergence quantity as intuitively as others. Although this shortcoming of $\sqrt{D_{\mathrm{JS}}}$ may not affect the deployment of $\sqrt{D_{\mathrm{JS}}}$ in numerical applications, it will hinder its deployment in applications of visual analytics, making it difficult to observe, analyze, and explain the relationships between a divergence value and its component measures and the contributions of different component measures.

We encountered this issue after we considered the synthetic cases in this section. In order to avoid the complication of introducing any synthetic case in the first part of the paper [4], we report this issue as an extra conceptual criterion in this second part of the paper. For this extra conceptual criterion, we give a 1 score to $\sqrt{D_{\mathrm{JS}}}$, and a 5 score to each of other five candidate measures.

## 6. Experimental Case Studies

To complement the synthetic case studies in Section 5, we conducted two surveys to collect some realistic examples that feature the use of knowledge in visualization. In addition to providing instances of criteria $R_{1}$ and $R_{2}$ for selecting a bounded measure, the surveys were also designed to demonstrate that one could use a few simple questions to estimate the cost–benefit of visualization in relation to individual users.

It is necessary to note that these surveys are not intended for evaluating any hypothesis related to the application concerned. They are designed to collect data that may be similar to the results of a controlled, semi-controlled, or uncontrolled empirical study, or to the estimation by a visual designer after an interview with potential users.

### 6.1. Volume Visualization (Criterion R${}_{1}$)

This survey, which involved ten surveyees, was designed to collect some real-world examples that reflect the use of knowledge in viewing volume visualization images. We invited surveyees with different levels of knowledge about volume visualization and medical imagining. They all volunteered their time as technical advisers without any financial reward. The full set of questions was presented to surveyees in the form of slides, which are included in the Appendix A.

The full set of survey results is given in Appendix C. The featured volume datasets were from “The Volume Library” [53], and visualization images were either rendered by the authors or from one of the four publications [54,55,56,57]. The transformation from a volumetric dataset to a volume-rendered image typically features a noticeable amount of alphabet compression.

Some major algorithmic functions in volume visualization, e.g., iso-surfacing, transfer function, and rendering integral, all facilitate alphabet compression, hence information loss. As a rendering integral, maximum intensity projection (MIP) incurs a huge amount of information loss in comparison with the commonly-used emission-and-absorption integral [58]. As shown in Figure 6, the surface of arteries are depicted more or less in the same color.

The accompanying question intends to tease out two pieces of knowledge, “curved surface” and “with wrinkles and bumps”. Among the ten surveyees, one selected the correct answer B, eight selected the relatively plausible answer A, and one selected the doubtful answer D. Among the participants, four rated their knowledge of medical imaging and volume visualization at 4 or 5 (out of 5). We consider them as an expert group. For this particular question (Figure 6), three selected answer A and one selected B.

Let alphabet Z={A,B,C,D} contain the four optional answers. Based on our observation of photographs online and consultation with medical doctors, we first assume a ground truth PMF $Q_{1}=\left\{0.1,0.878,0.002,0.02\right\}$ since there might still be a small probability for a section of an artery to be flat or smooth. The rendered image depicts a misleading impression, implying that answer C is correct or a misleading PMF $R_{C}=\left\{0,0,1,0\right\}$. The amount of alphabet compression is thus $H\left(Q_{1}\right)-H\left(R_{C}\right)=0.628$ bits.

The top four PCPs in Figure 7 show the measurements returned by the six candidate measures in a way similar to the PCPs in Figure 4 and Figure 5. In terms of **divergence ordering**, we notice a major anomaly that $D_{\mathrm{ncm}}^{k=1}$ returns divergence values indicating the “experts” group has the most divergence, followed by “all” and then “rest”. Looking at some marginal difference in detail, $D_{\mathrm{ncm}}^{k=2}$ indicates that “all” has the highest divergence, followed by “rest” and then “experts”. $D_{\mathrm{new}}^{k=1}$ indicates the group giving answer D has marginally more divergence than that giving answer C. These ordering conclusions are not intuitive. $D_{\mathrm{JS}}$, $\sqrt{D_{\mathrm{JS}}}$, and $D_{\mathrm{new}}^{k=2}$ returned the expected ordering, i.e., “rest” > “all” > “experts”, and C > D > A > B.

In terms of **benefit quantification**, $D_{\mathrm{JS}}$ and $\sqrt{D_{\mathrm{JS}}}$ suggest that “expert” is similar to making random guesses and “rest” is similar to the A group. $D_{\mathrm{ncm}}^{k}$ and $D_{\mathrm{new}}^{k}$ all consider that making random guesses is more beneficial than “expert”. This becomes a question about how to interpret the difference between $Q_{1}$ and {0.25,0.25,0.25,0.25}, and that between $Q_{1}$ and {0.75,0.25,0,0}, i.e., which is the more meaningful difference?

We thus introduce a second possible ground truth PMF based on the answers of “experts”, i.e., $Q_{2}=\left\{0.75,0.25,0,0\right\}$. The calculation results are depicted in the bottom four PCPs in Figure 7. In terms of **divergence order**, $D_{\mathrm{ncm}}^{k=1}$ shows an outlier, indicating the A group has more divergence than random guesses. With the observation of two PCPs in the first column of Figure 7, we consider $D_{\mathrm{JS}}$, $\sqrt{D_{\mathrm{JS}}}$, and $D_{\mathrm{new}}^{k=2}$ “excellent”, $D_{\mathrm{ncm}}^{k=2}$ and $D_{\mathrm{new}}^{k=1}$ “good”, and $D_{\mathrm{ncm}}^{k=1}$ “inadequate”.

$D_{\mathrm{JS}}$, $\sqrt{D_{\mathrm{JS}}}$, $D_{\mathrm{ncm}}^{k=1}$, and $D_{\mathrm{ncm}}^{k=2}$ all rate the C and D groups with the maximum divergence, while $D_{\mathrm{new}}^{k=1}$ and $D_{\mathrm{new}}^{k=2}$ do not. Following a careful reading of the intermediate calculation results, we notice that $D_{\mathrm{new}}^{k=1}$ and $D_{\mathrm{new}}^{k=2}$ would rate the divergence between {1,0,0,0} and {0,0,1,0} as the maximum divergence, but not for the divergence between {0.75,0.25,0,0} and {0,0,1,0}. This is an interesting feature of $D_{\mathrm{new}}^{k=1}$ and $D_{\mathrm{new}}^{k=2}$. We cannot decide whether to reward or penalize this feature. We hope to conduct future studies to examine the relative merits of this feature in detail.

In terms of **benefit quantification**, we cannot observe any major issues in the second column of Figure 7. We thus rate all candidate measures “excellent”.

From the PCPs in the third and fourth columns, we notice that with $Q_{2}$, more groups show a positive **impact of knowledge**. This is understandable, as $Q_{1}$ deviates more from the survey results. If we assume $Q_{1}$ is correct, then participants clearly do not have the necessary knowledge to answer the question in Figure 6 with the misleading MIP visualization. If $Q_{2}$ is correct, not only do the “experts” have the knowledge, but the “rest” group also seems to have some useful knowledge. In the $Q_{1}$-$K_{\psi }$ PCP, only $D_{\mathrm{JS}}$ and $\sqrt{D_{\mathrm{JS}}}$ indicate a positive knowledge impact for the “experts”. This is intuitive. In the $Q_{2}$-$K_{\psi }$ PCP, only $D_{\mathrm{new}}^{k=1}$ indicates a negative knowledge impact for the A group.

This is not intuitive. We thus consider $D_{\mathrm{JS}}$ and $\sqrt{D_{\mathrm{JS}}}$ “excellent”, $D_{\mathrm{ncm}}^{k=2}$, $D_{\mathrm{new}}^{k=1}$, and $D_{\mathrm{new}}^{k=2}$ “good”, and $D_{\mathrm{ncm}}^{k=1}$ “adequate”. By combining our evaluation of the ordering of divergence, benefit quantification, and the sign of knowledge impact, we give a 5 score to $D_{\mathrm{JS}}$, $\sqrt{D_{\mathrm{JS}}}$, a 4 to $D_{\mathrm{new}}^{k=2}$, a 3 to $D_{\mathrm{ncm}}^{k=2}$, and a 0 to $D_{\mathrm{ncm}}^{k=1}$.

### 6.2. London Underground Map (Criterion R${}_{2}$)

This survey was designed to collect some real-world data that reflects the use of knowledge in viewing different London underground maps. It involved sixteen surveyees, twelve at King’s College London (KCL) and four at University of Oxford. Surveyees were interviewed individually with the stimuli shown in Figure 1. Each surveyee was asked to answer 12 questions using either a geographically-faithful map or a deformed map, followed by two further questions about their familiarity of a metro system and London. A £5 Amazon voucher was offered to each surveyee as an appreciation of their effort and time. The survey sheets and the full set of survey results is given in Appendix D.

Harry Beck first introduced a geographically-deformed design of the London underground maps in 1931. Today, almost all metro maps around the world adopt this design concept. Information-theoretically, the transformation of a geographically-faithful map to such a geographically-deformed map causes a significant loss of information. Naturally, this affects some tasks more than others.

For example, the distances between stations on a deformed map are not as useful as in a faithful map. The first four questions in the survey asked surveyees to estimate how long it would take to walk (i) from *Charing Cross* to *Oxford Circus*, (ii) from *Temple* and *Leicester Square*, (iii) from *Stanmore* to *Edgware*, and (iv) from *South Rulslip* to *South Harrow*. On the deformed map, the distances between the four pairs of the stations are all about 50 mm. On the faithful map, the distances are (i) 21 mm, (ii) 14 mm, (iii) 31 mm, and (iv) 53 mm, respectively. According to the Google map, the estimated walk distances and times are (i) 0.9 miles, 20 min; (ii) 0.8 miles, 17 min; (iii) 1.6 miles, 32 min; and (iv) 2.2 miles, 45 min, respectively.

The average range of the estimations about the walk time by the 12 surveyees at KCL are: (i) 19.25 [8, 30], (ii) 19.67 [5, 30], (iii) 46.25 [10, 240], and (iv) 59.17 [20, 120] minutes. The estimations by the four surveyees at Oxford are: (i) 16.25 [15, 20], (ii) 10 [5, 15], (iii) 37.25 [25, 60], and (iv) 33.75 [20, 60] minutes. The values correlate better to the Google estimations than what would be implied by the similar distances on the deformed map. Clearly some surveyees were using some knowledge to make better inference.

Let Z be an alphabet of integers between 1 and 256. The range is chosen partly to cover the range of the answers in the survey, and partly to round up the maximum entropy Z to 8 bits. For each pair of stations, we can define a PMF using a skew normal distribution peaked at the Google estimation ξ. As an illustration, we coarsely approximate the PMF as $Q=\{q_{i}\;|\;1\leq i\leq 256\}$, where
$$q_{i}=\left\{\begin{array}{cc} 0.01/236 & \mathrm{if}\;\;1\leq i\leq \xi -8\;(\mathrm{wild}\mathrm{guess})\\ 0.026 & \mathrm{if}\;\;\xi -7\leq i\leq \xi -3\;\;\;\;(\mathrm{close})\\ 0.12 & \mathrm{if}\;\;\xi -2\leq i\leq \xi +2\;\;\;\;(\mathrm{spot}\mathrm{on})\\ 0.026 & \mathrm{if}\;\;\xi +3\leq i\leq \xi +12\;(\mathrm{close})\\ 0.01/236 & \mathrm{if}\;\;\xi +13\leq i\leq 256\;\;\;\;(\mathrm{wild}\mathrm{guess}) \end{array}\right.$$

Using the same way in the previous case study, we can estimate the divergence and the benefit of visualization for an answer in each range. Recall our observation of the phenomenon in Section 6.1 that the measurements by $D_{\mathrm{JS}}$, $\sqrt{D_{\mathrm{JS}}}$, $D_{\mathrm{new}}^{k=1}$, $D_{\mathrm{new}}^{k=2}$, $D_{\mathrm{ncm}}^{k=1}$ and $D_{\mathrm{ncm}}^{k=2}$ occupy different ranges of values, with $D_{\mathrm{new}}^{k=2}$ be the most generous in measuring the benefit of visualization. With the entropy of the alphabet as H(Q)≈3.6 bits and the maximum entropy being 8 bits, the benefit values obtained for this example exhibit a compelling pattern:
Benefit for:$D_{\mathrm{JS}}$$\sqrt{D_{\mathrm{JS}}}$$D_{\mathrm{new}}^{k=1}$$D_{\mathrm{new}}^{k=2}$$D_{\mathrm{ncm}}^{k=1}$$D_{\mathrm{ncm}}^{k=2}$*spot on*−1.765−2.777−0.418**0.287**−3.252−2.585*close*−3.266−3.608−0.439**0.033**−3.815−3.666*wild guess*−3.963−3.965−0.416−0.017−3.966−3.965

Only $D_{\mathrm{new}}^{k=2}$ has returned positive benefit values for *spot on* and *close* answers. Since it is not intuitive to say that those surveyees who gave good answers benefited from visualization negatively, clearly only the measurements returned by $D_{\mathrm{new}}^{k=2}$ are intuitive. In terms of **benefit quantification**, we consider thus $D_{\mathrm{new}}^{k=2}$ “excellent” and the other five measures “adequate”.

In addition, the ordering resulting from $D_{\mathrm{new}}^{k=1}$ is inconsistent with others. For **divergence order**, we consider $D_{\mathrm{new}}^{k=1}$ “adequate” and the other five measures “excellent”. We have not detected any major issues with the values for $K_{\upsilon }$ and $K_{\psi }$. For the **impact of knowledge**, we thus rate all candidate measures “excellent”.

More detailed discussions with further computational results and PCPs can be found in Appendix D. By combining all these observational ratings, we give a 5 score to $D_{\mathrm{new}}^{k=2}$, a 3 score to $D_{\mathrm{JS}}$, $\sqrt{D_{\mathrm{JS}}}$, $D_{\mathrm{ncm}}^{k=1}$, and $D_{\mathrm{ncm}}^{k=2}$, and a 1 score to $D_{\mathrm{new}}^{k=1}$.

## 7. Conclusions

This two-part paper aims to improve the mathematical formulation of an information-theoretic measure for analyzing the cost–benefit of visualization as well as other processes in a data intelligence workflow [3]. The concern about the original measure is its unbounded term based on the KL-divergence. The conceptual analysis in the first part of the paper [4] examined nine candidate measures and narrowed the options down to six, providing important evidence to the multi-criteria decision analysis (MCDA) of these candidate measures.

In the second part of the paper, we used two synthetic and two experimental case studies to obtain some data, which allowed us to observe the behaviors of the remaining candidate measures. Building on the MCDA in the first part, the case studies provided two additional aspects of the MCDA with important evidence.

From the top table in Table 1, we can observe that the empirical data helps identify the strengths and weaknesses of each candidate measures considered in this paper. The empirical data suggests that $D_{\mathrm{new}}^{k}\left(k=2\right)$ is slightly ahead of $\sqrt{D_{\mathrm{JS}}}$ (i.e., 16 vs. 15). Since the conceptual analysis in the first part of this paper [4] gives a subtotal of 30 to $D_{\mathrm{JS}}$, $\sqrt{D_{\mathrm{JS}}}$, and $D_{\mathrm{new}}^{k}\left(k=2\right)$. We cannot separate $D_{\mathrm{new}}^{k}\left(k=2\right)$ and $\sqrt{D_{\mathrm{JS}}}$ conclusively.

However, it is necessary to consider the extra conceptual criteria discussed in Section 5.3. From a visualization perspective, we cannot ignore the shortcoming of $\sqrt{D_{\mathrm{JS}}}$ discovered during the analysis of empirical data (i.e., its value is not the probabilistic mean of the entropic measures of its components). This places $D_{\mathrm{new}}^{k}\left(k=2\right)$ in a favorable position. We therefore propose to revise the original cost–benefit ratio in [3] to the following:(7)$$\begin{array}{cc} \frac{\mathrm{Benefit}}{\mathrm{Cos}t} & =\frac{\mathrm{Alphabet}\;\mathrm{Compression}-\mathrm{Potential}\;\mathrm{Distortion}}{\mathrm{Cos}t}\\ \; & =\frac{H\left(Z_{i}\right)-H\left(Z_{i+1}\right)-H_{\max}\left(Z_{i}\right)D_{\mathrm{new}}^{2}(Z_{i}^{\prime }\left|\right|Z_{i})}{\mathrm{Cos}t} \end{array}$$

This cost–benefit measure was developed in the field of visualization for optimizing visualization processes and visual analytics workflows. Its broad interpretation may include data intelligence workflows in other contexts [59]. The measure has now been improved by using visual analysis and with the empirical data collected in the context of visualization applications.

The history of measurement science [1] informs us that proposals for metrics, measures, and scales will continue to emerge in visualization, typically following the arrival of new theoretical understanding, new observational data, new measurement technology, and so on. As measurement is one of the driving forces in science and technology, we shall welcome such new measurement development in visualization.

The work presented in the first part of this paper [4] and this second part does not indicate a closed chapter but an early effort to be improved frequently in the future. For example, future work may discover measures that have better mathematical properties than $D_{\mathrm{new}}^{k=2}$, $\sqrt{D_{\mathrm{JS}}}$, and $D_{\mathrm{JS}}$, or future experimental observations may provide evidence that $D_{\mathrm{JS}}$ or $\sqrt{D_{\mathrm{JS}}}$ offer more intuitive explanations than $D_{\mathrm{new}}^{k=2}$ in other case studies. In particular, we would like to continue our theoretical investigation into the mathematical properties of $D_{\mathrm{new}}^{k}$.

“Measurement is not an end but a means in the process of description, differentiation, explanation, prediction, diagnosis, decision making, and the like” [7]. Having a bounded cost–benefit measure offers many new opportunities of developing tools for aiding the measurement and optimization of data intelligence workflows and for using such tools in practical applications, especially in visualization and visual analytics.

## Figures and Tables

**Figure 1 entropy-24-00282-f001:**
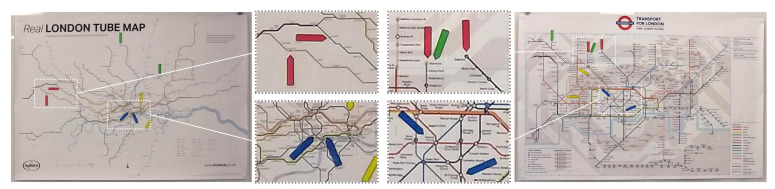
The London underground map (**right**) is a deformed map. In comparison with a relatively more faithful map **(left**), there is a significant amount of information loss due to many-to-one mappings in the deformed map, which omits some detailed variations among different connection routes between pairs of stations (e.g., distance and geometry). One common rationale is that the deformed map was designed for certain visualization tasks, which likely excluded the task for estimating the walking time between a pair of stations indicated by a pair of red or blue arrows. In one of our experiments, when asked to perform such tasks using the deformed map, some participants with little knowledge about London or London Underground performed these tasks well. Can information theory explain this phenomenon? Can we quantitatively measure relevant factors in this visualization process?

**Figure 2 entropy-24-00282-f002:**
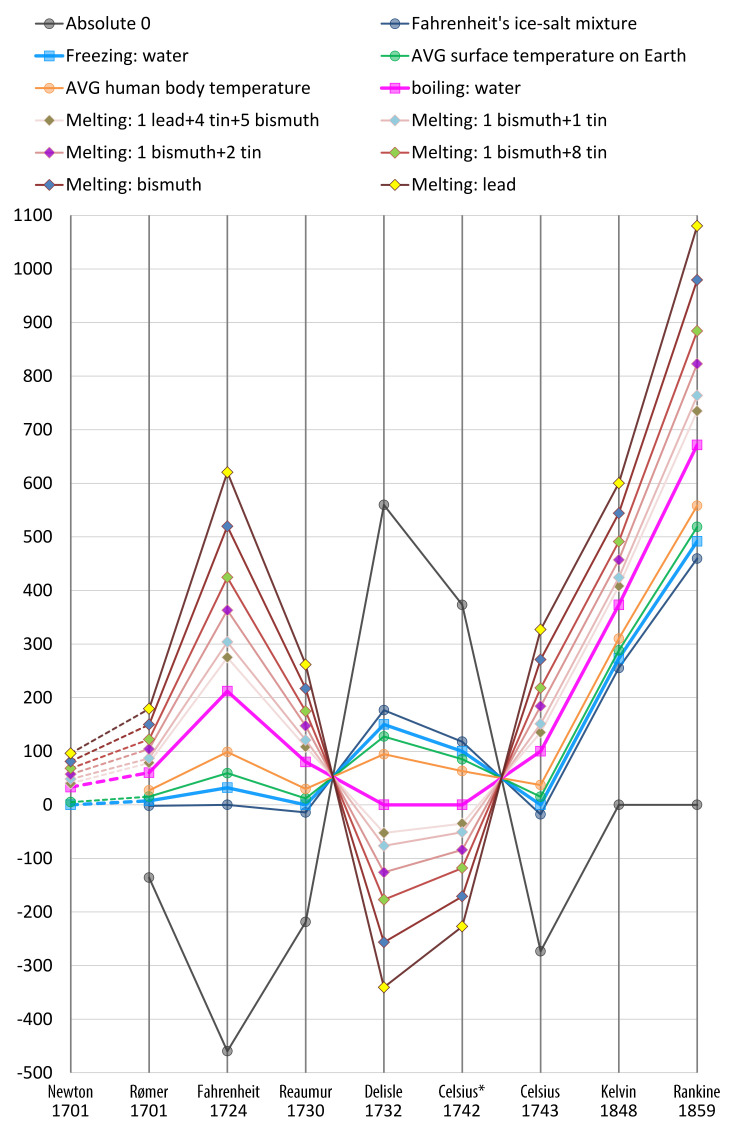
Major temperate scales proposed in history. Different lines show instances used as observation points, some of which became major reference points. Note: “Celsius* 1742” indicates the original scale proposed by Anders Celsius, while “Celsius 1743” indicates the revised Celsius scale used today that was proposed by Jean-Pierre Christin. The Newton scale is not linearly related to the others (shown as dash lines).

**Figure 3 entropy-24-00282-f003:**
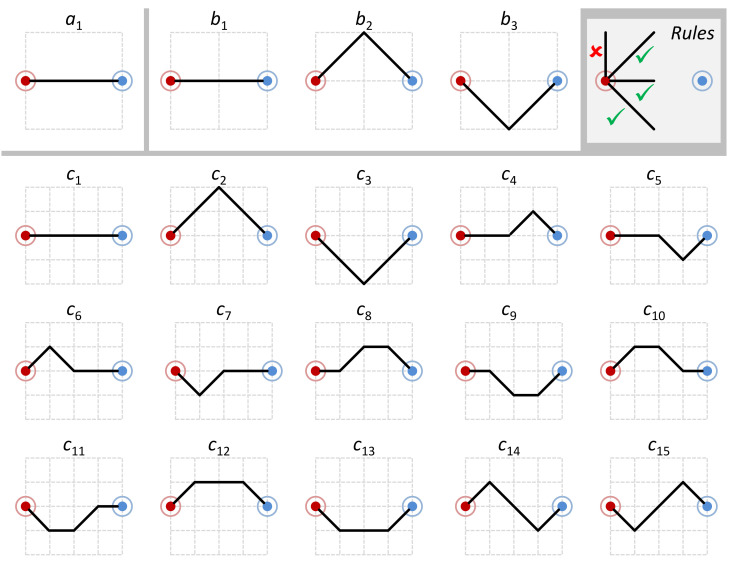
Three alphabets illustrate possible metro maps (letters) in different grid resolutions. Increasing the resolution enables the depiction of more reality, while reducing the resolution compels more abstraction.

**Figure 4 entropy-24-00282-f004:**
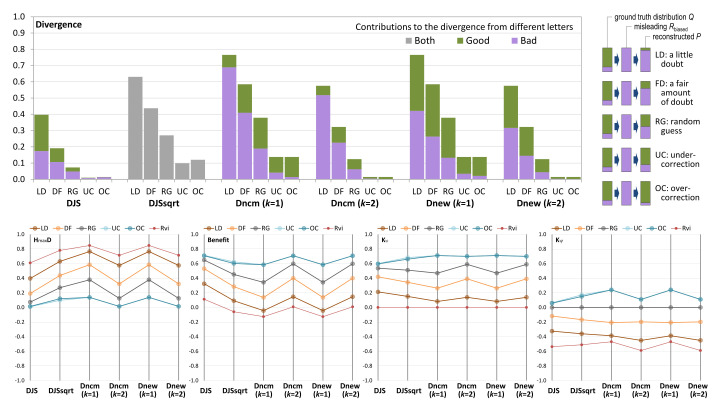
An example scenario with two states *good* and *bad* has a ground truth PMF Q={0.8,0.2}. From the output of a biased process that always informs users that the situation is *bad*. Five users, LD, DF, RG, UC, and OC, have different knowledge and thus different divergence. The five candidate measures return different values of divergence. We would like to see which sets of values are more intuitive. The illustration on the top-right shows two transformations of the alphabets and their PMFs, one by the misleading communication and the other by the reconstruction. The bar chart shows the divergence values calculated by each candidate measure, while the four parallel coordinate plots (PCPs) show the values of $H_{\max}D$ (divergence scaled by $SE_{\max}$), benefit, $K_{\upsilon }$ (impact of knowledge against relying solely on visual information), and $K_{\psi }$ (against random guess).

**Figure 5 entropy-24-00282-f005:**
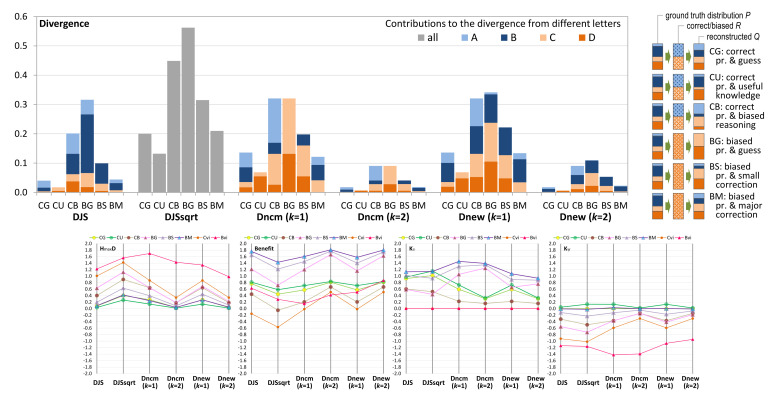
An example scenario with four data values: A, B, C, and D. Two processes (one correct and one biased) aggregated them to two values AB and CD. Users CG, CU, CB attempt to reconstruct [A, B, C, D] from the output [AB, CD] of the correct process, while BG, BS, and BM attempt to do so with the output from the biased processes. The bar chart shows the divergence values of the six users computed using the five candidate measures. The illustration on the right shows two transformations of the alphabets and their PMFs, one by the correct or biased process (pr.) and the other by the reconstruction. The bar chart shows the divergence values calculated by each candidate measure, while the four PCPs show the values of $H_{\max}D$ (i.e., divergence scaled by $SE_{\max}$), benefit, $K_{\upsilon }$ and $K_{\psi }$. The values for Cvi and Bvi correspond to $R_{\mathrm{correct}}$ and $R_{\mathrm{biased}}$, respectively.

**Figure 6 entropy-24-00282-f006:**
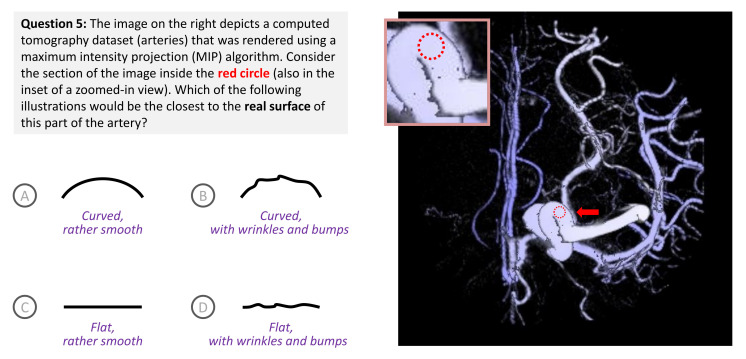
A volume dataset was rendered using the maximum intensity projection (MIP) method, which causes curved surfaces of arteries to appear rather flat. Posing a question about a “flat area” in the image can be used to tease out a viewer’s knowledge that is useful in a visualization process. This example was first described in Part I of this two-part paper [4] for demonstrating the role of human knowledge in dealing with information loss due to many-to-one mappings in such a visualization image. Similar to Figure 3 (Section 3) in this part, the example was used in Part I to illustrate the difficulty to interpret the unboundedness of the KL-divergence when considering a binary alphabet A={curved,flat} with maximum entropy of 1 bit.

**Figure 7 entropy-24-00282-f007:**
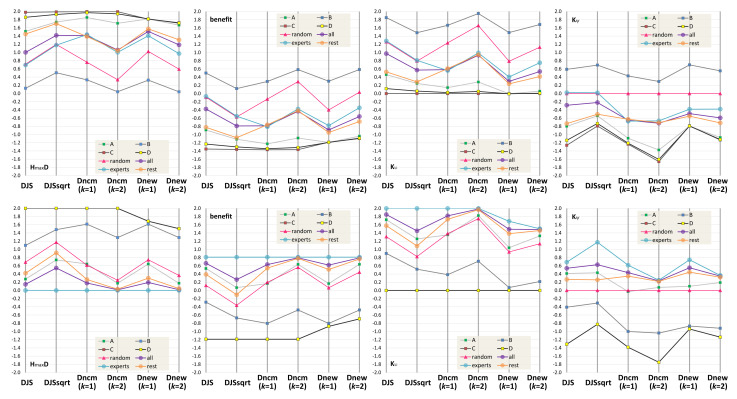
For the survey question shown in Figure 6, our survey of 10 participants returned 8 answers for A, 1 for B, 0 for C, and 1 for D. Among them, more knowledgeable participants (referred to as experts) returned 3 answers for A and 1 for B, and none for C or D. We consider two possible ground truth PMFs. $Q_{1}=\left\{0.1,0.878,0.002,0.02\right\}$ is based on our observations of photographs of arteries, and $Q_{2}=\left\{0.75,0.25,0.0,0.0\right\}$ is based on the experts’ survey results. The top four PCPs show the values of $H_{\max}D$, benefit, $K_{\upsilon }$, and $K_{\psi }$ calculated based on $Q_{1}$, while the bottom four PCPs are measured based on $Q_{2}$. In addition, we also consider five other groups that make a random guess or always answer A, B, C, or D.

**Table 1 entropy-24-00282-t001:** A summary of the multi-criteria decision analysis (MCDA). Each measure is scored against a criterion using an integer in [0, 5] with 5 being the best. Scores are calculated as: starting with a full score of 5. For each “good” deduct 1, each “adequate” deduct 2, and each “inadequate” deduct 3. The top table summarize the empirical scores obtained from the two synthetic case studies ($S_{1}$ and $S_{2}$) in Section 5 and two experimental case studies ($R_{1}$ and $R_{2}$) in Section 6. The bottom table presents the final results of MCDA by combining the subtotals of the seven conceptual criteria in the first part of the paper, the subtotals of the empirical criteria in this second part of the paper, and the scores of the extra conceptual criterion discussed in Section 5.3.

A Summary of the Empirical Scores Obtained of the Four Case Studies							
**Criteria**		$D_{\mathrm{JS}}$	$\sqrt{D_{\mathrm{JS}}}$	$D_{\mathrm{new}}^{k=1}$	$D_{\mathrm{new}}^{k=2}$	$D_{\mathrm{ncm}}^{k=1}$	$D_{\mathrm{ncm}}^{k=2}$
**S_1_:**	*order*	adequate	adequate	excellent	excellent	good	good
	*benefit*	adequate	good	excellent	good	excellent	good
	*knowledge*	excellent	excellent	excellent	excellent	excellent	excellent
	score	1	2	5	4	4	3
$S_{2}:$	*order*	excellent	excellent	good	excellent	adequate	good
	*benefit*	adequate	excellent	good	adequate	good	adequate
	*knowledge*	excellent	excellent	excellent	excellent	excellent	excellent
	score	3	5	3	3	2	2
$R_{1}:$	*order*	excellent	excellent	good	excellent	inadequate	good
	*benefit*	excellent	excellent	excellent	excellent	excellent	excellent
	*knowledge*	excellent	excellent	good	good	adequate	good
	score	5	5	3	4	0	3
$R_{2}:$	*order*	excellent	excellent	excellent	excellent	adequate	excellent
	*benefit*	adequate	adequate	adequate	excellent	adequate	adequate
	*knowledge*	excellent	excellent	excellent	excellent	excellent	excellent
	score	3	3	3	5	1	3
Empirical Subtotal:		12	15	14	16	7	11
**Combining All Scores Obtained from the Conceptual and Empirical Evaluation**							
**Criteria**		$D_{\mathrm{JS}}$	$\sqrt{D_{\mathrm{JS}}}$	$D_{\mathrm{new}}^{k=1}$	$D_{\mathrm{new}}^{k=2}$	$D_{\mathrm{ncm}}^{k=1}$	$D_{\mathrm{ncm}}^{k=2}$
Conceptual Subtotal [4]:		30	30	28	30	26	29
Empirical Subtotal:		12	15	14	16	7	11
Componentization (extra criterion):		5	1	5	5	5	5
**Total without the extra criterion**:		**42**	**45**	**42**	**46**	**33**	**40**
**Total with the extra criterion**:		**47**	**46**	**47**	**51**	**38**	**45**

## Data Availability

The survey results are reported in Appendix C and Appendix D.

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
