# Peer review of "A Bounded Measure for Estimating the Benefit of Visualization (Part II): Case Studies and Empirical Evaluation"

_entropy, 2022, doi:10.3390/e24020282_

Round 1

Reviewer 1 Report

This is mostly a well-written well-thought-out paper. Most of my comments relate to typos or English. The one real issue I have is with Fig 6. I don't understand how the lines participants were asked to draw relate to the image.

The text
 14 is yet conclusive
probably should be
 14 is not yet conclusive

The word "indicate" should be "indicates" in this line
 56 amount of actual distortion is lower than the supposed distortion, this indicate that the

This line:
 72 There is currently no standard measurement scales for measuring
Should either be this:
 72 There is currently no standard measurement scale for measuring
or this:
 72 There are currently no standard measurement scales for measuring

There should be a space after as in this line:
 101 different data points at his proposed scale has been considered as“the first attempt

Possibly "we do not need these..."?
 262 we do not these data points to specify a scale

Something went wrong here. Not sure what you're trying to say.
 286 those users, who do not have much knowledge and make random guesses, would lead
a uniform PMF Puniform. Those users, who do not have much knowledge and reason
 287 to
 288 about the options in

"Prefect" should probably be "perfect" below:
 290 the prefect combination between the available visual

"anyway" should be "any way".
 339 good-related divergence. This asymmetric pattern is not in anyway incorrect,

The close quote on the word excellent should be a double quote.
 344 terms of ordering, we consider Dk new “excellent’,

I don't understand Fig 6. The inside of the red circle seems to be utterly featureless to me. What is meant by surface? I don't see how that relates to any of the lines, A through D.

Something is off with this sentence. Either a word is missing or maybe the word "integral" should be removed.
 437 In terms of rendering integral, maximum

Possibly this should be "Looking for some..."?
 457 Looking some marginal difference in detail

Possibly "can" should be "cannot" in this line.
 480 In terms of benefit quantification, we can observe any major issues

Possibly "that" should be "then" in this line.
 484 more from the survey results. If we assume Q1 is correct, that participants

This line is missing the words "do" and "but" which I have inserted.
 486 MIP visualization. If Q2 is correct, not only [do] the “experts” have the knowledge, [but] the “rest”

This line is missing the word "a" which I have inserted
 505 Harry Beck first introduced [a] geographically-deformed design

The word "distance" should be "distances."
 516 and (iv) 53mm respectively. According to the Google map, the estimated walk distance

The word "time" should be "times."
 517 and time are

Missing space between "of" and "benefit"
 536 In terms ofbenefit quantification

The text
 565 technology, and so on. As measurement is one
should be
 565 technology, and so on as measurement is one

The word "development" should be "developments" in this line.
 566 technology. We shall welcome such new measurement development in visualization

The word "provides" is missing in line 570. Also, "observation" should be "observations". The text
 570 ... or future experimental observation may evidence that DJS or p
should be
 570 ... or future experimental observations may provide evidence that DJS or p

Author Response

See the attached PDF file.

Reviewer 2 Report

I think there are many publishable results in this paper, but the presentation is confusing, the nomenclature is somewhat sloppy, there appear to be oversights or typos, and the English needs editing.

For the presentation, it is confusing since the manuscript implies that this is essentially two papers but one is in the Appendix?  The manuscript needs to be split into two papers and the important appendix needs to be submitted and published to set up the current results.

Second, the use of the term "Information" is ambiguous.  Information appears in the first sentence of the Abstract without a clear meaning.   Shannon's Information Theory is very explicit in its definition but this manuscript is not.  To be considered a scientific contribution, this terminology must be clarified. 

Third, the adoption of the term "Alphabet Compression" is a bit confusing to me.  Of course, compression is a well-defined part of information theory; however, alphabet compression as used here is really just alphabet combining--nothing exotic going on is there?

Fourth, a native English speaker needs to read the entire manuscript and clean up poor English usage.

Author Response

See the attached PDF file.
